# Tornado Risk Climatology in Europe

**Jürgen Grieser * and Phil Haines ***

RMS, Peninsular House, London EC3 8NB, UK
* Correspondence: juergen.grieser@rms.com (J.G.); phil.haines@rms.com (P.H.);
  Tel.: +44-79-1251-9404 (J.G.); +44-20-7444-7803 (P.H.)

**Abstract:** Violent tornadoes are rare in Europe but they can have devastating effects. Damage associated with individual tornadoes can reach several billion euros and they have caused hundreds of fatalities. The tornado risk varies considerably over Europe, but so far only a few national maps of tornado risk and one Europe-wide map exist. We show several different ways to create quantitative maps of tornado occurrence rates as follows: Kernel smoothing of observations, climatologies of convective parameters from reanalysis, output of a logistic regression model to link convective parameters with observed tornadoes, orography-dependent climatologies and finally the population-bias corrected tornado occurrence rates from the Risk Management Solutions (RMS) Europe Severe Convective Storm Model. We discuss advantages and disadvantages of each approach and compare the results. While the climatologies created from the individual methods show a lot of qualitative similarities, we advocate to combine the methods to achieve the most reliable quantitative climatology.

**Keywords:** tornado; risk climatology; Europe

## 1. Introduction

Tornadoes are rare, small but potentially dangerous [1]. Maybe the deadliest tornado in Europe in the modern era was the one that hit a paper factory in Monville, France, on 19 August 1845 with at least 80 fatalities [2]. According to Meaden [3], in the year 1091 a tornado flattened about 600 wooden-frame houses in central London, and central Paris got hit by a tornado on 10 September 1890. If a violent tornado were to hit a European urban centre today, it could cause several billion euros of damage. Even though hundreds of tornadoes occur every year in Europe, the tornado risk can be seen as high-impact but low-frequency since only a small portion of tornadoes reach devastating intensities. Risks of this type are usually underestimated [1]. They disappear from collective memory before the next event occurs, as discussed for major floods by Fanta et al. [4]. It is important to quantify such risks, especially for the insurance industry who should understand all drivers of risk to their portfolio.

Risk climatologies are an essential tool to quantify local threats. For some countries of Europe national tornado risk climatologies already exist. These include the United Kingdom [5], Germany [6], Poland [7], Austria [8], Greece [9], Spain [10] and Italy [11]. However, spatial and temporal variations in population density and reporting practices make direct comparison difficult. Groenemeijer and Kühne (2014) [12] published a Europe-wide tornado climatology created by kernel smoothing of all recorded tornadoes from the European Severe Weather Database (ESWD) [13]. While this risk climatology is Europe wide, it too suffers from reporting biases and does not quantify tornado occurrence rates. In this paper we give examples of several different ways to create quantitative Europe-wide tornado climatologies, discuss the advantages and disadvantages of each approach and compare the results. The goal is to provide a quantitative tornado risk map providing local occurrence rates as the expected number of tornadoes occurring per 100 years and per 10,000 km$^2$ while minimising the impact of variations in reporting practices and population density.

In Section 2.1 we discuss kernel smoothing of observed tornado locations. We first give results for the U.S. to show how this method, together with additional information, can be used to estimate an intensity-specific hit rate for any location. We then give results for Europe and discuss why this approach does not lead to reliable quantitative results. In Section 2.2 we show tornado climatologies based on convective parameters calculated from reanalysis data. How such parameters can be combined with local observations and one example of a climatology following from this is shown in Section 2.3. Far more tornadoes are observed in flat and low lands than in the rough mountainous areas of Europe. In Section 2.4 we show how this information can be used to create quantitative tornado risk climatologies. Finally, in Section 2.5 we describe how the various approaches discussed in previous sections are used by Risk Management Solutions (RMS) to create a quantitative view of tornado risk for 15 countries in Europe.

## 2. Different Routes to a European Tornado Climatology

### 2.1. Kernel Smoothing

A simple way to create a tornado risk map is to do so directly from historical observations. In this section we estimate a local occurrence rate of tornadoes by applying kernel smoothing to historical reports. We start with an example from the U.S. in order to (a) show how much can be gained and (b) better understand the limitations of kernel smoothing when then applied to Europe.

The Storm Prediction Centre (SPC) of the U.S. collects tornado reports since 1950 in a national database [14] based on reports from National Weather Service forecast offices. The data are collected and quality controlled following strict policies from the National Centers of Environmental Information (former National Climatic Data Center, NCDC) and got reviewed by the NCDC in 1988 [15]. Verbout et al. (2006) [16] discussed potential quality issues and highlight that there is no trend in the annual number of tornadoes with intensity F1 or greater on the Fujita scale. According to the Pacific Northwest National Laboratory, the SPC database is in reasonably good condition for the kind of analysis performed here [17]. It has been used for similar analyses before, e.g., by Brooks et al. (2003) [18].

For each tornado, the SPC database provides (among other information) the location, the path length, maximum width and maximum intensity using the Fujita (F) scale [19]. Over the first few decades the number of recorded tornadoes shows a significant trend. Since 1991 no trend is visible and the number of tornadoes recorded in the U.S. can be seen as stationary. We apply a Gaussian kernel smoother to the locations of all 34,133 tornadoes recorded between 1991 and 2018 to obtain a tornado occurrence rate density $\rho$ as an expected number of tornadoes per period (100 years) and per reference area (10,000 km$^2$), see Figure 1a.

Feuerstein et al. [20] showed that the tornado intensity distribution (the fraction of tornadoes that reach a certain intensity) does not vary significantly between different data collections covering different regions of the world. Dotzek et al. [21] explained this with an exponential distribution of the maximum kinetic energy within a tornado. Using their intensity distribution $\alpha_F$ allows transforming the results of the kernel smoothing of all tornadoes in a region into an occurrence rate density $\rho_F = \alpha_F \rho$ per intensity class based on the F scale. Only 188 of the recorded tornadoes since 1991 reach an intensity of F4 or F5. Kernel smoothing of only these violent tornadoes would lead to a far less representative spatial distribution than kernel smoothing all observed tornadoes and then applying the constant fractions of violent to all tornadoes discussed in [20]. This follows directly from the small number of observations of violent tornadoes.

The tornado occurrence rate density by intensity ($\rho_F$) is only the first step of a local risk assessment. In order to estimate a climatology of potential damage caused by tornadoes, we need to know how often, on average, a subject at risk in an individual location is "hit" with intensity $I$. This can be expressed as a local hit rate $h_I$ defined as hits with intensity $I$ per time. Assuming local spatial homogeneity, the hit rate is identical to the area fraction that is hit with intensity $I$ within a reference

area. We can calculate $h_I$ using two further pieces of information. For each tornado intensity class F we need to know both the average footprint area $A_F$ and then, for each intensity $I \leq F$, the average fraction $a_{F,I}$ of the footprint that experiences a maximum intensity of $I$

$$h_I = \sum_{F=I}^{5} a_{F,I}\, \rho_F A_F = \rho \sum_{F=I}^{5} a_{F,I}\, \alpha_F A_F. \tag{1}$$

Both, $A_F$ and $a_{F,I}$ are estimated from observations as we discuss now.

Brooks (2003) [22] showed that the average tornado footprint area $A_F$ increases with increasing intensity and modelled the intensity-specific distributions of tornado footprint length and maximum width with Weibull distributions. Elsner et al. (2014) [23] refined the link between tornado footprint dimensions and intensity from a discrete F scale to a continuous windspeed scale. Here, we use the SPC data themselves to estimate the footprint area of each recorded tornado. We use the length and maximum width provided by SPC and approximate tornado footprints by ellipses. Knowing the average footprint size and the local occurrence rate of tornadoes allows us to calculate a location-specific rate for experiencing a tornado of a given intensity class.

Tornado intensity is expressed by the maximum intensity a tornado develops during its lifetime. Standohar-Alfano and Lindt (2015) [24] analysed the footprints of several tornadoes to estimate the area fractions $a_{F,I}$ of a footprint with weaker than maximum intensity expressed in the F scale. Fricker et al. [25] confirmed and generalised these results. As an example, the average footprint area in which an F5 tornado reaches F5 intensity is only about 2% of its total footprint area. We use these results and Equation (1) to estimate the local hit rate with given intensity. The result are tornado risk climatologies for the U.S. by intensity. These risk climatologies can be combined to obtain the risk of a location hit with intensity $I$ or greater, indicated as $I^+$ by

$$h_{I+} = \sum_{i=I}^{5} \sum_{F=i}^{5} a_{F,i}\, \rho_F A_F = \rho \sum_{i=I}^{5} \sum_{F=i}^{5} a_{F,i}\, \alpha_F A_F. \tag{2}$$

As an example we show the local hit rate with intensity of F3 or greater in Figure 1b. Note that the two maps in Figure 1 look identical since according to Equation (2) the local hit rate depends linearly on the local tornado occurrence rate. In other words, if a region sees twice as many tornadoes then the chance that a location is hit with F3 or stronger intensity is also twice as high. Representative vulnerability functions for specific subjects at risk like cars, houses, wind turbines or nuclear power plants can be used to translate the local hit rate into specific damage ratios, and to estimate the return periods of a given damage ratio or even complete destruction.

In Europe, the European Severe Storms Laboratory (ESSL) was founded in 2002. It operates the European Severe Weather Database (ESWD) which is designed to collect severe weather data covering Europe [13]. Although the ESSL has made efforts to include historic reports, most of the reports in the ESWD are from more recent years. This has led to an artificial trend in the number of recorded tornadoes. Like the SPC, the ESSL has strict reporting criteria that clearly distinguish between tornadoes, gustnadoes, funnel clouds, dust devils and downbursts. Data included in the database can have four quality levels which are as received (QC0), plausibility checked (QC0+), confirmed by reliable source (QC1) and fully verified (QC2). Anyone can report a tornado sighting. Tornadoes reported or verified by a trained spotter or a national hydro-meteorological service get the quality level QC0+. QC1 is given to sightings verified by the ESSL and QC2 is only given to tornadoes for which a scientific case study with detailed information exists. During the period 2005–2019 only three reported tornadoes in the ESWD have quality level QC0, 25.2% have QC0+, 69.4% have QC1 and 5.4% have QC2. We therefore expect that the database is not contaminated with reports of other meteorological phenomena and is adequate to be used for the analyses we perform in this paper. There is, however, the issue of underreporting which is discussed in Section 2.5.

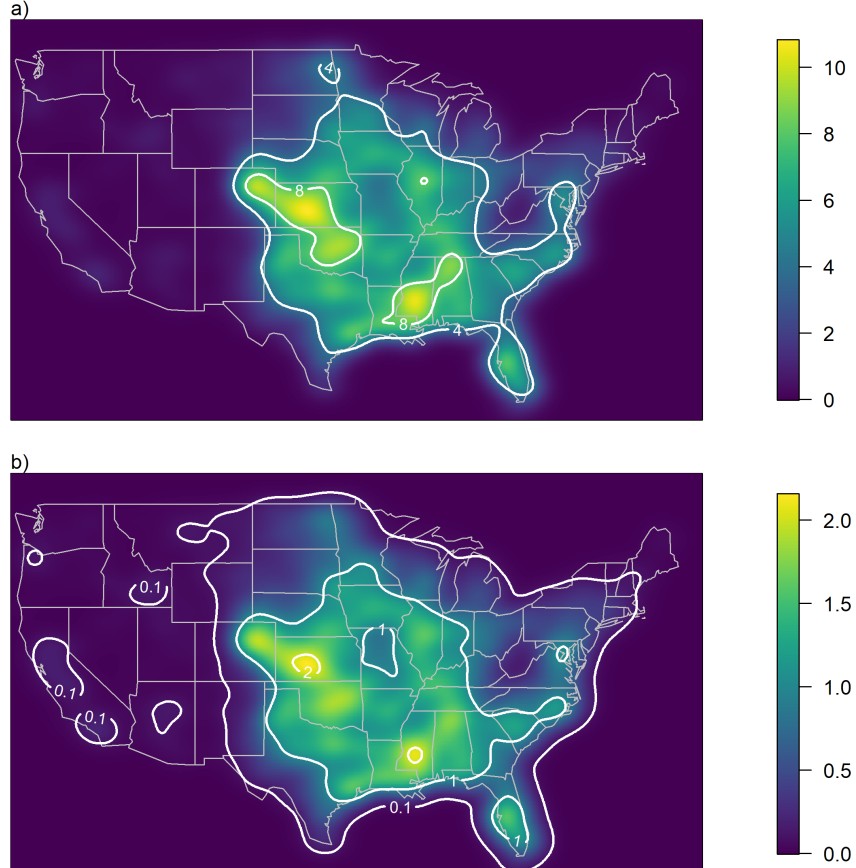

**Figure 1.** Tornado Climatology of the United States: (**a**) tornado occurrence rate per year and per 10,000 km$^2$ based on observations from 1991 to 2018, (**b**) local hit rate per 100,000 years with intensity of at least F3. While (**a**) answers the question how often tornadoes occur in a region, (**b**) answers the question how often on average a location in a region is hit with at least F3 intensity (for details see text). Contours of 4 and 8 tornadoes per year and 10,000 km$^2$ are drawn in map (**a**) and contours of 0.1, 1 and 2 hits per 100,000 years are drawn in map (**b**) as well.

Groenemeijer and Kühne (2014) [12] published a Europe-wide tornado climatology created by kernel smoothing of all recorded tornadoes from the ESWD. As they discussed, the climatology is impacted by a population bias as well as a shortage of information from some countries. Since they used all observations available rather than restricting to a specific time period their results cannot be used to obtain an accurate occurrence rate.

Several additional years of dedicated tornado observations in Europe are now available. This provides us with the opportunity to apply kernel smoothing including these observations and estimate local occurrence rates from it. We limit our analysis to the area bounded by the longitude range from −11 to 32 degrees and the latitude range from 33 to 72 degrees. The ESWD contains 12,419 tornado records within this region, the locations of which are depicted in Figure 2a. Applying the Mann–Kendall trend test, for the 15 years between 2005 and 2019 we see no significant trend in the number of tornadoes recorded per year. During this period 6481 tornadoes are recorded (see Figure 2b) which is just over half of the total number for this region.

Since we are interested in tornado risk we do not want to count waterspouts that never reach land. We therefore exclude waterspouts that never make landfall by dropping database entries for which both SURFACE_INITIAL_LOCATION and SURFACE_CROSSED are set to one of SEA, LAKE or WATER which leaves 3792 tornadoes (see Figure 2c). More than 41% of the recorded tornadoes since 2005 are in fact waterspouts that never make landfall. The fraction of recorded tornadoes that are pure

waterspouts is generally higher in countries with long coastlines but can also be high in land-locked countries. In Switzerland, two third of the recorded tornadoes are actually waterspouts, occurring over lakes without making landfall. Only 1821 out of the 3792 recorded tornadoes that touched land in the period 2005–2019 have an intensity rating (see Figure 2d). This is far less than for the U.S. where all the recorded tornadoes have an F rating. These numbers suggest that with an average of 1219 tornadoes a year in the contiguous U.S. compared to 252 tornadoes in Europe, the U.S. reports around 5 times more tornadoes than Europe. Note that we used our bounding box to define Europe with a land area of 6.5 million km$^2$ compared to the area of the contiguous U.S. of 7.6 million km$^2$.

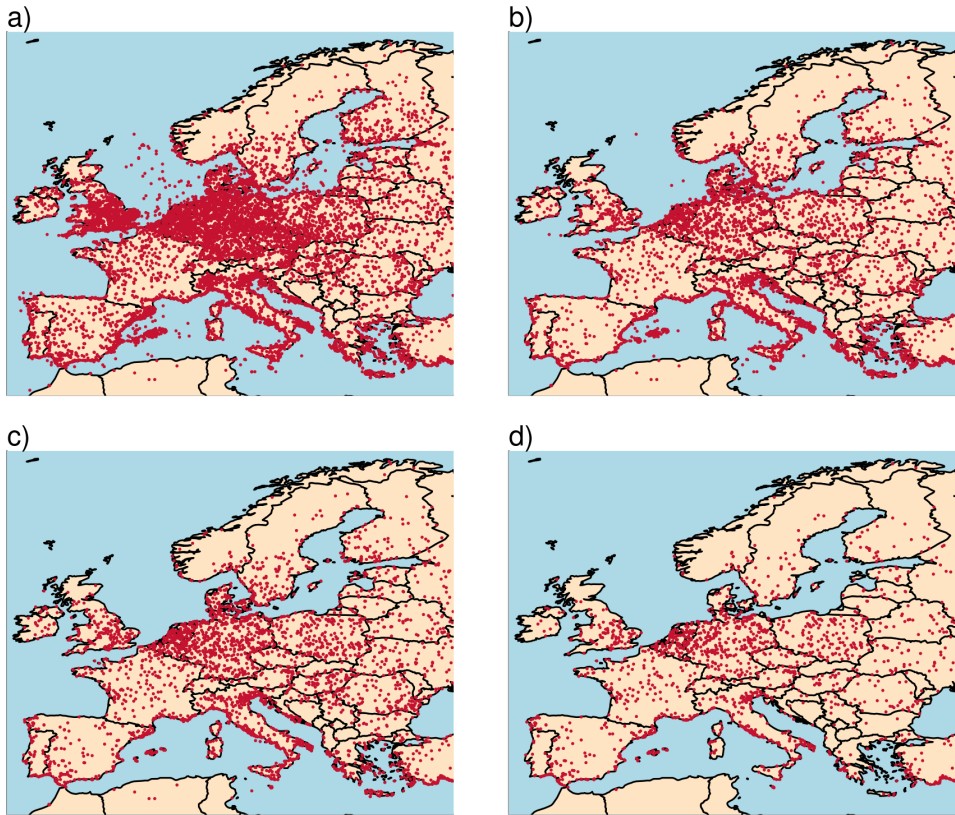

**Figure 2.** Locations of European Severe Weather Database (ESWD) tornado reports: (**a**) all recorded tornadoes, (**b**) tornadoes recorded in the period 2005–2019, (**c**) tornadoes recorded in the period 2005–2019 that touched land and (**d**) tornadoes recorded in the period 2005–2019 that touched land and have an intensity rating.

We apply kernel smoothing to the 4 different datasets we just discussed, i.e., using (a) all recorded tornadoes, (b) tornadoes recorded in the period 2005–2019, (c) tornadoes recorded in the period 2005–2019 that touched land and (d) tornadoes recorded in the period 2005–2019 that touched land and have an intensity rating. The resulting climatologies are depicted in Figure 3 as occurrence rate per 100 years and per 10,000 km$^2$. In case (a), where no period is determined, we use the ratio of total number of observations to the observations within the 15 years from 2005 to 2019 to normalise the number of observations. Therefore, Figure 3a,b have the same area-wide tornado occurrence rate.

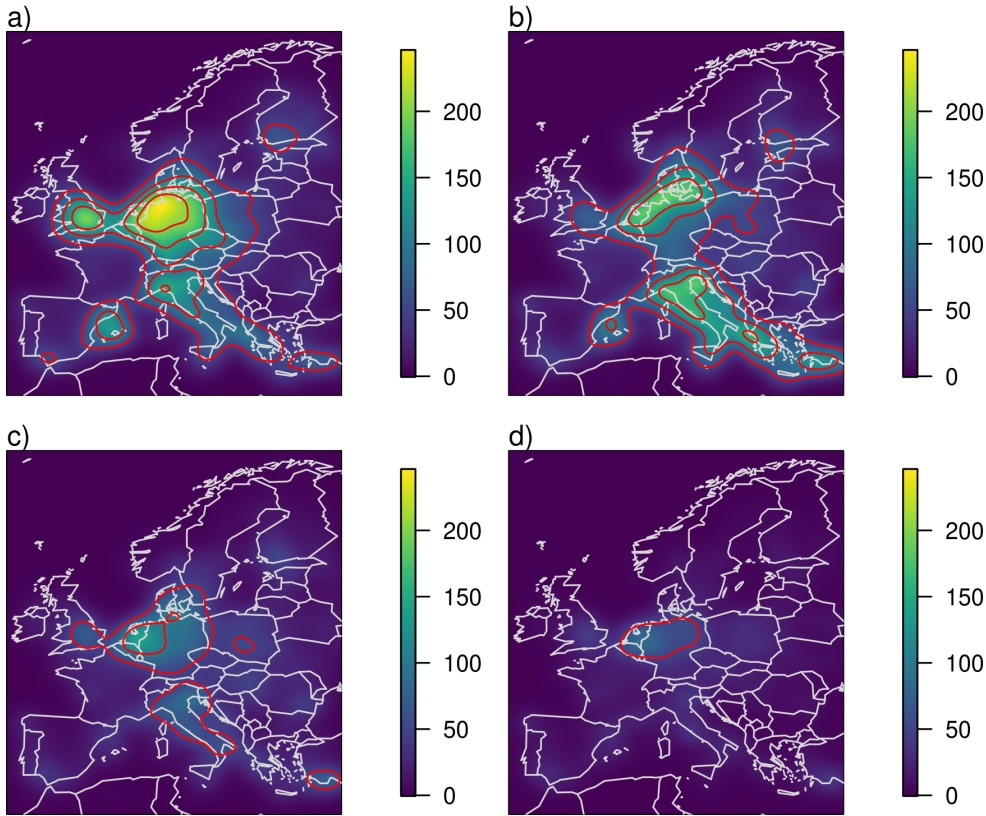

**Figure 3.** Kernel smoothed ESWD tornado reports per 100 years and 10,000 km$^2$ based on (**a**) all recorded tornadoes, (**b**) tornadoes recorded in the period 2005–2019, (**c**) tornadoes recorded in the period 2005–2019 that touched land and (**d**) tornadoes recorded in the period 2005–2019 that touched land and have an intensity rating. Contour lines of 50, 100, 150 and 200 reports per 100 years and 10,000 km$^2$ are drawn in red.

We now discuss the results of kernel smoothing of reported tornado locations in Europe. The countries and regions mentioned in the text can be found on the map in Appendix A (Figure A1). It is not surprising that the result when all recorded tornadoes are used shows many similarities with the map produced by Groenemeijer and Kühne [12]. They show the same high tornado occurrence in northern Germany, Belgium and the Netherlands, southern England and the north of Italy. More pronounced is the high occurrence count over the Balearic Islands reaching to the mainland coast of Spain. In order to see which part of this climatology is driven by the temporal change in observations, we compare it to the results we obtain when we restrict the data to the stationary period from 2005 to 2019 in Figure 3b. In this case the local maximum over northern Germany, Belgium and the Netherlands becomes smaller, less pronounced and more aligned with the flat lands near the coast. We also see high occurrence rates over all of Italy, parts of the Greek coast and reaching south to the Turkish coast. The local maxima over southern England and the Balearic Islands are still there but less pronounced. The differences between the two climatologies can easily be explained by the national ratios of tornadoes recorded within the recent 15 year period to the total number of recorded tornadoes. While it is nearly 70% for Italy (most of the overall tornado observations there are from recent years), it is only 21% for the United Kingdom and 32% for Germany. In the last two countries there is a dedicated effort to find and analyse historical evidence of tornado occurrences. In the UK this is done for many years by the Tornado Research Organization (TORRO) [5]. In Germany this work was pioneered by Wegner [26] and Letzmann [27].

Waterspouts that never reach land contribute more than 40% of the total tornado count in the ESWD within the period from 2005 to 2019 and contribute substantially to the maxima observed in Figure 3b. When interested in the risk to humans and property on land we need to exclude them from the analysis. This is done in case (c) and the resulting climatology is shown in Figure 3c. The highest occurrence rates are now visible over the flat areas of the Netherlands, northern Belgium and north-western Germany. A second maximum is in the Po Valley in northern Italy and a third local maximum can be seen along the south coast of Turkey. The local maximum over the Balearic Islands is smaller than in the other two cases although parts of the southern Spanish coast still show enhanced occurrence rates, as does the whole of Italy.

Figure 3d shows the occurrence rate of intensity-rated tornadoes. A comparison to Figure 3c shows where people make the additional effort to rate the intensity of tornadoes which can be very difficult, especially if tornadoes do not hit buildings as discussed e.g., in [13].

When Gaussian kernel smoothing is applied, a two-dimensional Gaussian bell is centred around each observation to obtain a smoothly interpolated occurrence rate. Such a Gaussian bell has three degrees of freedom which can be expressed as the variances or bandwidths in x and y directions and the correlation. In all our applications we set the correlation to zero and used the suggested (optimal) settings for the bandwidth from Venables and Ripley [28] with respect to case a, i.e., the use of 12,419 locations. This means that we used the same length scales for the interpolation of all four cases which allows us to compare the spatial structures undisturbed by different smoothing length scales. We also did the interpolations with the optimal bandwidth for each case. It turned out that in some cases local information gets lost. Especially in Italy where most of the tornadoes occur either at the Adriatic or Mediterranean coast, a larger bandwidth leads to a maximum in the inland of Italy and lower occurrence rates at the coast. This reflects a major disadvantage of kernel smoothing, it does not consider any meteorological or orographic effects. In the following section we discuss approaches that are fully independent of individual observations of tornadoes and solely based on meteorological environments.

## 2.2. Meteorological Environments

An alternative way of creating a convective risk climatology, independent of observations, is to use reanalysis data. Riemann et al. [29] provide global climatologies of convective available potential energy (CAPE) and convective inhibition (CIN), both of which are necessary ingredients for the development of severe weather. Prein and Holland [30] published a global hail climatology based on a combination of convective parameters optimised using observations from the U.S. Brooks et al. [31] combined several parameters including mixed layer (ML) CAPE and 1km vertical wind shear to produce a coarse-resolution global tornado climatology.

Several parameters are developed in weather forecasting to describe the risk of tornadoes. Although these parameters are designed for conditional forecasting of severity, they can also be calculated from reanalysis data to create tornado risk maps independent from actual observations. Among them are the Craven parameter [32], which is the product of MLCAPE and 0–6 km shear, and the significant tornado parameter (STP) [33]. In this study we use an alternative, surface layer, definition of STP suggested by the SPC. This uses surface based (SB) rather than mixed layer CAPE and is defined as

$$STP = \frac{SBCAPE}{1500\,Jkg^{-1}} \frac{2000 - SBLCL}{1000\,m} \frac{SRH_{1km}}{150\,m^2s^{-2}} \frac{S_{6km}}{20\,ms^{-1}} \frac{200 + SBCIN}{150\,Jkg^{-1}}.$$

Here SBLCL denotes the lifted condensation level (LCL) associated with a surface based parcel, $SRH_{1km}$ is the $0-1$ km storm relative helicity and $S_{6km}$ is the surface to 6 km wind shear. The SBLCL and SBCIN terms are bounded between 0 and 1. The shear term is bounded above by 1.5 and set to zero for $S_{6km} < 12.5\,m\ s^{-1}$.

We calculate values of the Craven parameter and STP using climate forecast system reanalysis (CFSR) data [34,35] between 1979 and 2015. For each day and 0.5° grid cell we take the maximum value of the parameter across the 8 output times, and then aggregate over the 13,514 individual days to obtain a climatology. Figure 4 shows the total number of days for which each parameter exceeds a threshold suggested by the original authors [32,33]. For the Craven parameter we use 30,000 m$^3$ s$^{-3}$, given as a possible lower threshold for significant tornadoes. For STP we choose 1, which was found to maximise skill in discriminating between significantly tornadic and non-tornadic supercells.

Figure 4 shows that, with the exception of a few grid cells at the Mediterranean coast, neither threshold is exceeded regularly in Europe. There are a large number of grid cells for which the thresholds are never exceeded during the 37 year study period. We recognise that these thresholds were designed to be used with U.S. soundings, however, we reproduce the original thresholds here to illustrate that parameters and associated thresholds developed for use in a specific region may not be directly applicable in another region. It is also clear from Figure 4 that the exceedance of these thresholds can differ dramatically between neighbouring grid cells, but caution is advised here given the extreme values involved. Finally we observe that, although the parameters contain similar ingredients, the spatial distribution of risk that they imply is dependent on the exact way in which these ingredients are combined. In the next section, we fit a statistical model to these and many other parameters to establish which combinations can best predict the occurrence of historic tornadoes in Europe.

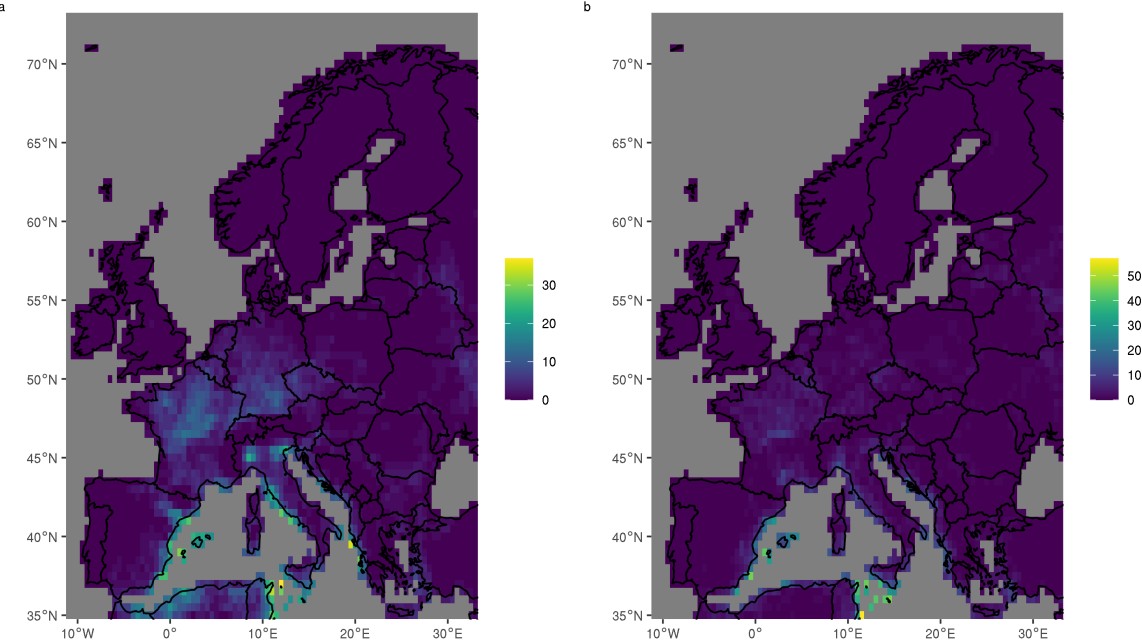

**Figure 4.** Count of climate forecast system (CFS) reanalysis days between 1979 and 2015 with (**a**) Craven parameter greater than 30,000 m$^3$ s$^{-3}$ and (**b**) significant tornado parameter greater than 1.

### 2.3. Combination of Observations and Meteorological Environments

The meteorological ingredients of convection can be linked with observations of severe weather by using a statistical model to map historical values of convective parameters to an occurrence probability. This model can be trained on a subset of the data for which there are reliable observations and then applied to a larger data set to produce a climatology. Previous examples for Europe include Mohr et al. [36,37] who used a logistic regression to produce a climatology of hail, and Rädler et al. [38] who used an additive logistic regression to produce climatologies of lightning, hail and wind.

Using CFS reanalysis data we calculated a large number of parameters with a potential link to the occurrence of tornadoes. These parameters, all of which are associated with the ingredients of convection, can be loosely separated into three groups. Firstly, those that contain values of temperature and humidity at only a few levels in the atmosphere, including lifted indices. Secondly, those that are primarily measures of wind shear and lastly, a more complex class of parameters associated with a vertical integral of the atmosphere, such as CAPE. This third class additionally includes those parameters that combine several others. Two such parameters, Craven and STP, were discussed in the previous section. We also introduce one additional parameter in this group, defined as the product of convective available potential energy integrated only over the lowest 3 km of the atmosphere, $SBCAPE_{3km}$, and wind shear between the surface and 1 km, $S_{1km}$

$$S1C3 = S_{1km}SBCAPE_{3km}.$$

As previously reported by Manzato [39], there is a high degree of correlation between many of the indices within each group, and to a lesser extent between the first and the third group.

We apply a logistic regression model to link daily values of these convective parameters to ESWD reports of tornado occurrence. The ESWD reports are filtered to remove non-landfalling waterspouts as in Section 2.1 and then each report is assigned to the CFS grid cell in which it occurred. We initially use tornado reports from Germany and Italy between 2006 and 2015, years for which there is not a large trend.

Logistic regression [40] uses the logit link function $g(\mu) = \log(\mu/(1 - \mu))$ to specify the probability of occurrence, $\mu$, of a random variable in terms of a linear combination of $p$ predictor variables $x_j$

$$g(\mu) = \beta_0 + \beta_1 x_1 + \ldots + \beta_p x_p$$

where the coefficients $\beta_j$ of the linear combination are to be estimated using the maximum likelihood approach. In order to establish how well each parameter can explain the occurrence of historic tornadoes in the ESWD database we first fit a regression to each one individually. The most effective parameter, measured by the likelihood of the fitted model, is selected, and the process is then repeated stepwise to add further parameters. This process resulted in a two parameter fit to the modified K index ($K_m$) [41] and to the new composite parameter $S1C3$, with the following form

$$g(\mu) = -14.7 + 0.178K_m + 0.00133\,S1C3 \tag{3}$$

For illustrative values $K_m$ = 40 °C and S1C3 = 4000 m$^3$ s$^{-3}$, (3) gives a probability of tornado occurrence of approximately 0.1. Figure 5 shows a climatology of tornado occurrence obtained by applying (3) to values for the whole of Europe between 1979 and 2015. The daily probabilities output from the logistic regression model are aggregated over the entire period, and then rescaled to give a tornado occurrence rate per 100 years and 10,000 km$^2$.

The highest tornado occurrence rates in Figure 5 are found at the Mediterranean coast, notably for Italy and parts of the Adriatic. High values are also found in the East towards the Russian border, although we caution that this region is far from the area used to train the model and also well outside the intended region of application (see Section 2.5). Moderately high values are found in the Po Valley, and over large areas of northern France and Benelux. Lower values are found for most of Scandinavia and the Iberian peninsula as well as over mountain ranges. The high values over northern continental Europe are in line with the results from Groenemeijer and Kühne [12] who argue that gradients within countries may be less susceptible to reporting biases, and note a decrease in tornado reports from the northwest to the southeast for both France and Germany. This gradient of tornado activity can be contrasted with the finding of other authors that convective activity in general, as measured by lightning [42,43] and overshooting cloud tops [44], decreases to the north and west across Germany and France.

The high values over northwestern Europe seen in Figure 5 are due to the inclusion of S1C3 in the regression. Average values of $SBCAPE_{3km}$ are found to have a very different spatial pattern than other measures of CAPE. We find that the mean of $SBCAPE_{3km}$ is driven by small ($<100 \, J \, kg^{-1}$) amounts of CAPE, which are frequently found across northwestern Europe. In contrast, the mean of SBCAPE is driven by less frequent, but much larger, values that are found typically on either side of the Alps as well as along the Mediterranean coast. Comparing to $SBCAPE_{3km}$ alone the additional inclusion of shear in S1C3 leads to a further decrease in average values in the Mediterranean relative to those in northwestern Europe.

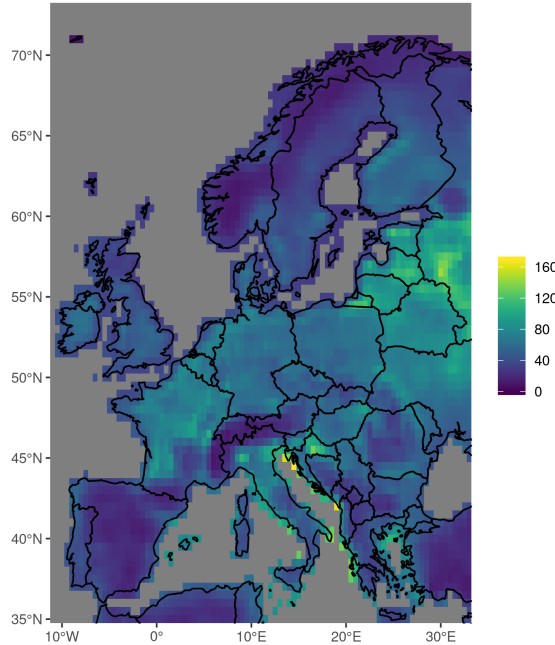

**Figure 5.** Tornado occurrence rate per 100 years and 10,000 $km^2$ obtained from the output of the two-parameter logistic regression model given by (3).

With so many highly correlated parameters it is not surprising that the exact combination chosen is sensitive to the data used to train the model. In order to investigate this sensitivity we examined a large number of alternate fits using either a subset of years (between 2006 and 2015) or a different subset of countries (from Austria, Germany, Italy and Switzerland). We make the following general observations. The modified K index, lifted indices in general and $SBCAPE_{3km}$ (both with and without the additional shear multiple) all performed consistently well. In every case considered they were either selected as the first parameter, or came close to being selected (had a comparable likelihood to the selected parameter). Other measures of CAPE performed markedly less well, and were not close to being selected even as the second or third parameter.

Several previous studies have found lifted indices to outperform other parameters for predicting severe convection in Europe [39,45,46]. However, the same studies also report considerable skill for CAPE. We note three differences between these studies and our own which may offer an explanation for the poor performance of CAPE here. Firstly these studies all used soundings data rather than reanalysis. The exact magnitude of CAPE in individual cases may not be well represented by reanalysis data. Previous studies have found a large degree of scatter when comparing calculations of CAPE from reanalysis with those from soundings, both in Europe [47,48] and elsewhere [49,50]. Secondly these studies all used CAPE above a threshold as a discriminant, making them less sensitive to the exact magnitude of CAPE. It is possible that beyond a certain threshold the relationship of CAPE to tornado occurrence departs significantly from the linear dependence our model assumes. Westermayer et al. [51] found that the frequency of lightning in Europe increased with increasing

CAPE only up until around 800 J kg$^{-1}$. Finally none of these studies were focussed on tornadoes. The success of SBCAPE$_{3km}$ here suggests that CAPE at low levels is of more importance than overall CAPE for tornado formation. CAPE below 3 km has been previously found to distinguish tornadic environments from other forms of severe convection both in the U.S. [52] and in Europe [53,54].

Although shear is clearly important for tornado formation, direct measures of shear were never selected as one of the first three parameters. Again this may be due to the model assumption of a linear relationship that is independent of the other parameters, a limitation of the method employed here. Shear was often included indirectly either via S1C3 or the MCS maintenance probability (MMP) [55] which is itself the output of a logistic regression. We note the caution of Doswell and Schultz [56] with regards to combining indices in arbitrary ways. However, the success of S1C3 suggests that the interaction of SBCAPE$_{3km}$ and shear is important in a way that cannot be captured by their additive combination alone, and implies that future work to explore this interaction more fully might be of value.

## 2.4. Orographic Influence

Jagger et al. (2015) [57] found that spatial variations in the occurrence frequency of observed tornadoes in Kansas (U. S.) are related to elevation roughness and provide a statistical model describing this relationship. Elsner et al. (2016) [58] examined this relation further by varying the definition of elevation roughness, grid resolution and other variables. The European tornado climatologies we provided so far, based on observations only, reanalysis only and a combination of reanalysis and observations, show spatial patterns that follow orography. Therefore, we now take a different approach and consider the distribution of tornado occurrence as a function of orographic characteristics only.

We use orographic data published by Earth Environment [59] and analyse various features like elevation, slope and terrain roughness. As Amatulli et al. (2018) [59] have shown, there is some correlation between these different orographic features. We attribute orographic characteristics to each observed tornado occurrence based on the Earth Environment data with a spatial resolution of 0.008333 degrees or approximately 1km. As in Section 2.1, we distinguish four cases and visualise the tornado occurrence distribution as a function of elevation in the form of exceedance probability (EP) curves. Exceedance probabilities are 1 minus the cumulative density function (cdf) and are useful for examining the tail of a distribution.

We estimate the EP curves from the observations using order statistics (see e.g., [60]) and the median estimate of the cdf value for each observation. Order statistics also allows for estimates of the sampling uncertainty. We show the estimated EP curves for all four cases and the 90% confidence interval for case (c) in Figure 6. Cases (a) and (b) show an accumulation of tornado observations at an elevation of 392m which is visible in the sharp drop of the EP curves. They originate from the fact that 56 out of 58 tornadoes recorded over Lake Constance are actually waterspouts which did not make landfall.

From Figure 6, it is clear that the number of observed tornadoes decreases rapidly with elevation. Half the tornadoes observed in the period 2005–2019 occurred in elevations lower than 89 m. The average tornado elevation is 155 m.

In addition, shown in this Figure is the EP curve of area as a function of elevation within the analysed region. If there was no statistical dependence between elevation and tornado occurrence, the EP curves of both, area and tornado occurrence, would not be significantly different. However, the difference is massive. While about 10% of the area is above 1500 m less than one in 1000 tornadoes is observed in or above that elevation.

We can use the two EP curves to estimate the tornado occurrence rate as a function of elevation. To minimise the sampling error, we do not use equidistant elevation bands since this can lead to a small number of tornadoes in individual bands. Instead, we define them by decentiles of the number of tornadoes. We can apply this strategy to any orographic variable. Figure 7 shows the resulting tornado

climatologies for mean elevation, slope, terrain roughness index and sub-grid standard deviation of elevation for grid cells of 10 km.

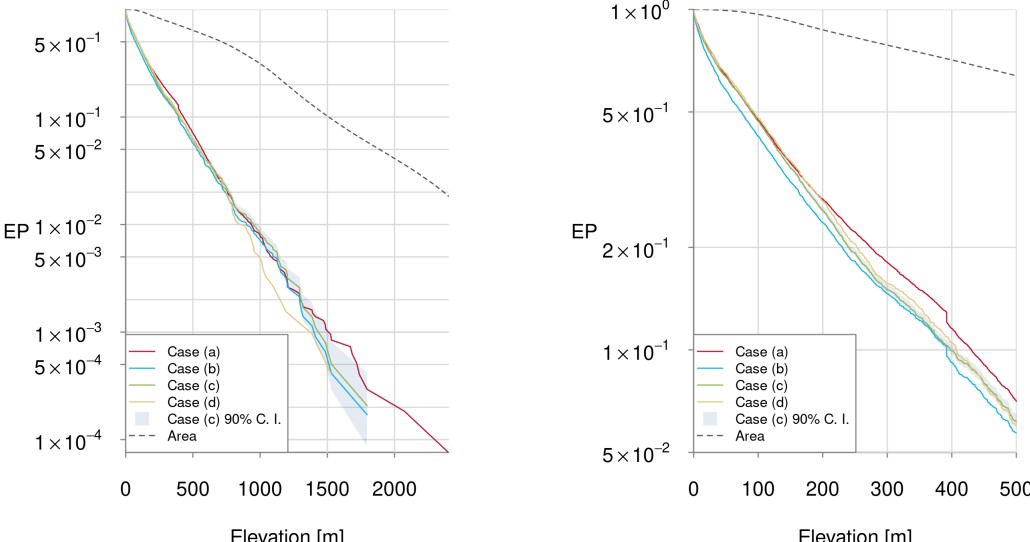

**Figure 6.** Exceedance probability curves of tornado observations for 4 cases: (a) all recorded tornadoes (red line), (b) tornadoes recorded in the period 2005–2019 (blue line), (c) tornadoes recorded in the period 2005–2019 that touched land (green line) and (d) tornadoes recorded in the period 2005–2019 that touched land and have an intensity rating (yellow line). Right graph is restricted to elevations below 500 m for better visibility. The 90% confidence interval is drawn for case (c) as shaded region, tornadoes that hit land in the period 2005–2019. The exceedance probability (EP) curve for the distribution of area over elevation is also shown by a dashed line.

The four maps share a lot of similarities illustrating the high degree of correlation between the orographic variables within the analysis region. This means that the clear statistical dependence of tornado occurrence visible in Figure 6 might in fact camouflage a dependence on other orographic features like slope or roughness. There are also some differences between the maps. If pure elevation is used, very high tornado occurrence rates are more concentrated in areas with very low elevation.

Nevertheless, all four maps also show features that became visible by kernel smoothing of the observed tornado locations. While kernel smoothing interpolates observed occurrence densities symmetrically in all directions, the method of this section interpolates solely with respect to orographic features. All maps show local maxima along the German, Dutch, Belgian and British coast of the North Sea. A pronounced maximum occurs in the Po Valley in northeastern Italy. A small local maximum is visible at the south coast of Spain. All these features are visible in Figure 3 as well.

Since no climatological information and no local observations are used, some features are clearly unrealistic. Lakes, as the Ladoga Lake in Russia, salt plains in Algeria and Tunisia and the flat lands along the coast of the North Sea all have the same low slope, sub-grid standard deviation of elevation and terrain roughness. The method of this section attributes the same tornado occurrence rate to all of them. While the link to orographic features introduces small-scale variability which seems to be more realistic than kernel smoothing, it does not yield reliable results if applied across different climate zones.

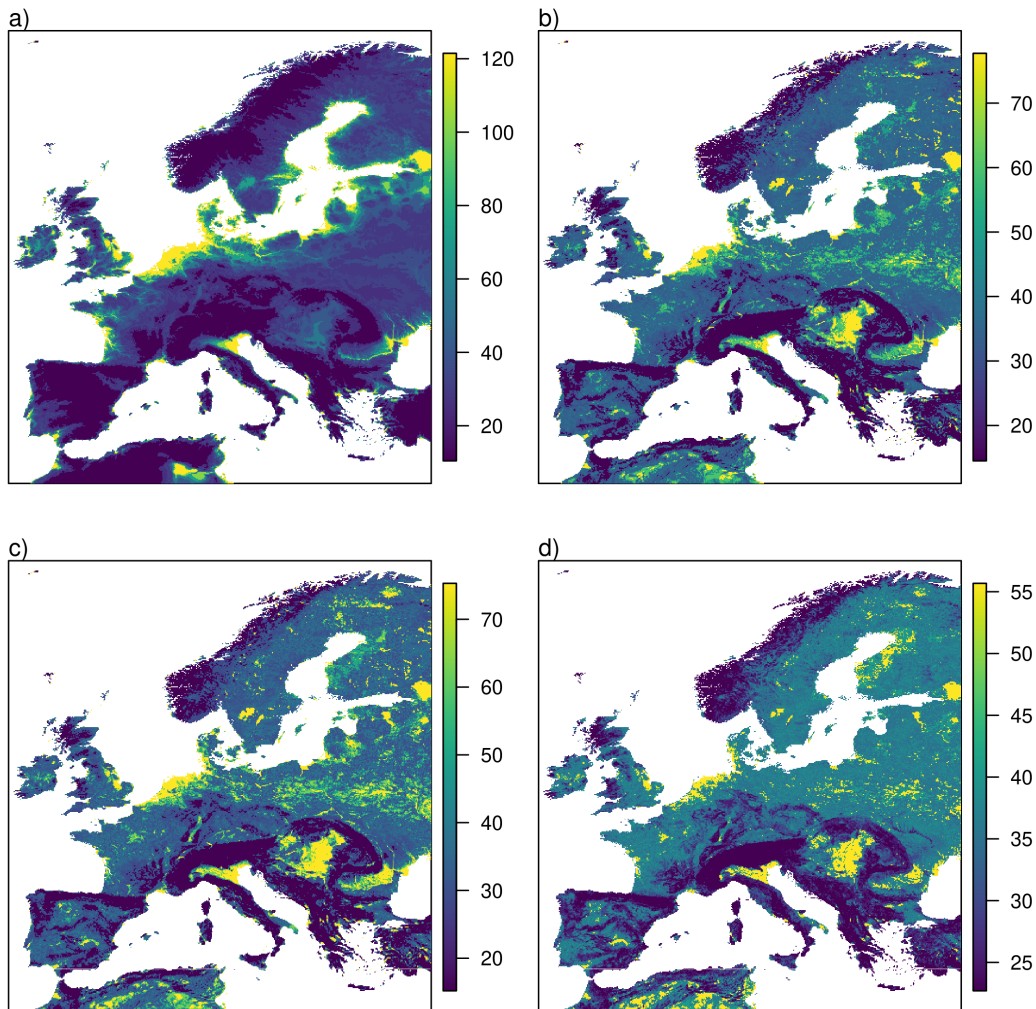

**Figure 7.** Local tornado occurrence rate per 100 years and 10,000 km$^2$ derived solely from orographic characteristics on 10km resolution: (**a**) elevation, (**b**) slope, (**c**) terrain roughness index and (**d**) sub-grid standard deviation of elevation.

### 2.5. A Tornado Risk Climatology

Each of the methods discussed in the previous sections has its specific advantages and disadvantages. Kernel smoothing is based on observations but suffers from different reporting practices in different countries. The use of convective parameters is independent of local reporting practices but suffers from coarse resolution and results that depend on the specific choice of ingredients. Convective parameters are also sensitive to the reanalysis data used to calculate them as are statistical models that rely on those parameters, by association. Methods that are solely based on orographic information describe the clearly visible dependence of tornado occurrence rate on orographic conditions but neglect information about spatial variations in the meteorological conditions that are relevant to tornado occurrence. We now describe how these approaches can be combined to create a tornado risk climatology that benefits from the advantages of each method. A climatology created in this way is included in the new RMS Europe Severe Convective Storm model, which covers 15 countries.

In a first step, non-landfalling waterspouts and tornadoes associated with extra-tropical cyclones are excluded from the ESWD tornado reports. In a second step we impose the total number of tornadoes per year for the entire region in order to address under reporting. Our estimate for the total number of tornadoes within the region is calculated by applying the following population-bias correction. The population bias results from the fact that in less populated areas fewer tornadoes can

be observed than in denser populated areas (see e.g., [57] or [61]). To obtain an estimate of the total number of tornadoes from the number of observed tornadoes per country, we selected a base country with a relatively high tornado and population density and a relatively large area to minimise sampling uncertainty. These criteria are fulfilled by Germany. The observed tornado density for other countries is then divided by the ratio of the population density of the country and the population density of Germany for those countries with a lower population density than Germany. This bias correction increased the number of tornadoes per year in the domain from 135 to 181. The highest correction factor was found for Ireland (3.4) followed by Austria (2.3) and Slovakia (2.0). Our estimate of 181 tornadoes per year in the domain compares well to the estimate from a survey among national experts done at the European Conference on Severe Storms of the year 2002 which led to 186 to 194 tornadoes (Luxembourg and Liechtenstein excluded) [62]. Note, that we exclude tornadoes spawned by extratropical storms. Our approach does not take into account biases independent of population density, however, this good agreement supports the assumption that other biases (e.g., lack of interest or technical infrastructure) might be less important during the period from 2005 to 2019.

We then fit a logistic regression model, using the approach described in Section 2.3, to connect the CFS reanalysis data to the tornado reports allowing us to produce a probability of tornado occurrence for each day between 1979 and 2015 and for each 0.5° grid cell. We do so using non bias corrected tornado reports, since we have no way to assign under reported tornadoes to specific locations or dates. We show the results in Figure 8a rescaled so that 181 tornadoes per year are generated on average in the domain.

The output of this regression is sensitive to the spatial pattern of instability in the CFSR input data. Previous studies have noted differences in the spatial pattern of severe convection over Europe both between different models [63] and between models and soundings [47]. Groenemeijer et al. [64] found pronounced differences in the distribution of CAPE values over 1000 J kg$^{-1}$ between CFS and ERA-Interim [65] reanalysis data. Therefore, as an additional step intended to make the results less dependent on the reanalysis product used, we apply local corrections to our tornado occurrence probabilities in two regions where the CFS and ERA reanalyses differ. The areas identified for local corrections are Puglia in the South-East of Italy and Veneto in the North-East of Italy. In the first area we reduced probabilities by up to 20%, in the latter by up to 10%. We did so by overlaying Gaussian bells over the spatial probability distributions. While this procedure is subjective, it dampens local extrema originating from the use of CFSR data which are not supported by observations and ERA data. Figure 8b shows the probability map from the logistic regression after local corrections are applied.

We also smooth the values between neighbouring CFSR grid cells to dampen some of the more extreme variation between neighbouring grid points in the reanalysis noted in Section 2.2. The smoothed map is shown in Figure 8c. To allow for variation at scales finer than can be resolved by the reanalysis resolution of 0.5° we downscale to 0.1° by applying a local elevation weighting based on the results discussed in Section 2.4. The daily local tornado occurrence probabilities are then aggregated to obtain a tornado risk map which is again rescaled to produce an average of 181 tornadoes per year across the domain. This final RMS tornado risk map is shown in Figure 8d.

We created this climatology based on tornado favouring weather conditions, using a logistic regression model to link the most reliable tornado reports to large scale features found in the reanalysis data. This allows what is learnt from reports to be applied uniformly across the domain, avoiding artificial drops at national borders such as the one visible between Belgium and France in Figure 2. Features of the logistic regression model (Figure 8a) that are preserved in the final climatology include the high rate of tornado occurrence at the Italian coast and in the Po Valley, as well as the relatively high values across northwestern France, the last of which is also found by Groenemeijer and Kühne [12]. The evident link between orography and tornado occurrence is applied locally allowing us to capture the fine scale elevation effects seen in Figure 7. For example lower occurrence rates in the Italian Apennine mountains are a feature that was not well resolved by either kernel smoothing of reports or by the reanalysis data. By introducing the elevation dependence locally we avoid equating all regions

with the same elevation. Whereas in Figure 7 many other low lying coastal regions share the same high tornado occurrence rate as the Po Valley in northern Italy, in Figure 8d they do not.

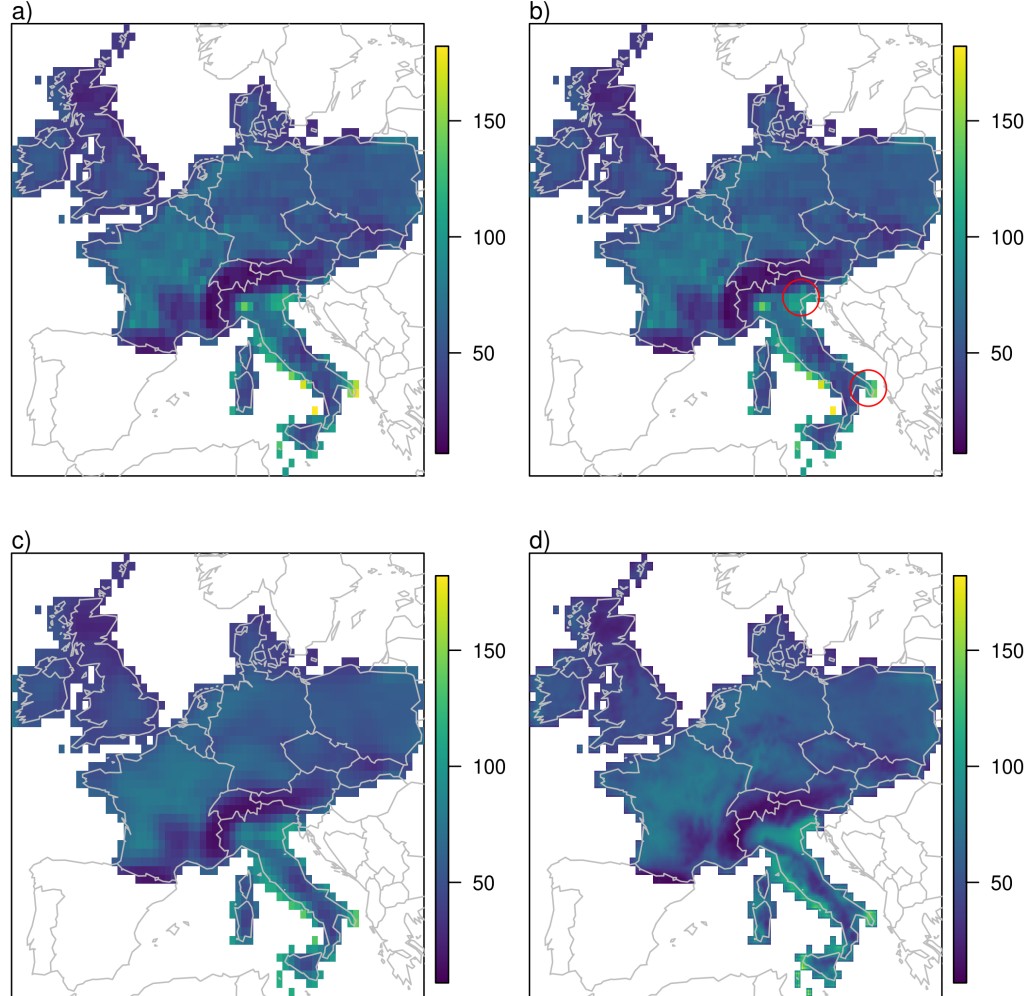

**Figure 8.** Local tornado occurrence rate of the Risk Management Solutions (RMS) Severe Convective Storms Model per 100 years and 10,000 km$^2$. In all four cases the rates are scaled to produce an average of 181 tornadoes per year across the entire domain. Panel (**a**) rescaled output of the GLM, (**b**) including corrections at two locations where the frequency of $SBCAPE$ >1000 J kg$^{-1}$ in climate forecast system reanalysis (CFSR) seems unrealistically high compared to ERA, (**c**) following smoothing between neighbouring grid cells and (**d**) final occurrence rate after downscaled to 0.1° using an elevation weighting. Red circles highlight the two areas where local corrections are applied in panel (**b**).

A climatology of tornado risk such as is presented in Figure 8d is only one component of a catastrophe model. In Section 2.1 we showed how information about the occurrence rate of tornadoes can be converted into a local hit rate by intensity, which is required, together with information about the vulnerability of subjects at risk, in order to estimate the expected average annual loss to a location. This approach, which relies on the assumption that neither the average size or average intensity of tornadoes varies spatially, allows information that is very valuable to regional authorities and the insurance industry to be calculated directly from a climatology of tornado occurrence. However, this still lacks information about spatio-temporal correlation. Local occurrence rates do not provide information about the size of individual footprints and certainly not of whole events. For example, tornado outbreaks, which can produce extraordinary high losses exceeding ten billion USD. This is a

general shortcoming of risk climatologies which can only be avoided with detailed risk modelling of individual outbreaks, such as is done by catastrophe-modelling companies.

Results also depend on the accuracy of the observations. Grieser and Terenzi [66] performed a Monte-Carlo simulation to estimate how the uncertainty of various components translates into national economic loss ratios.

## 3. Discussion, Conclusions, and Outlook

In this paper we demonstrated several methods that can be used to produce a tornado climatology for Europe. Each of these methods has advantages and disadvantages. Kernel smoothing is a simple method that can be applied successfully and can be combined with other data such as average footprint size or the fraction of violent tornadoes to all tornadoes to deliver additional insights. We applied the method to the U.S. where more than 1200 tornadoes are observed per year, obtaining similar results to Brooks et al. [18]. In Europe, where the number of recorded tornadoes per year is far lower and the period for which observations have been systematically collected is much shorter, the results depend strongly on the filtering of observations. While it is a general disadvantage of kernel smoothing that it assumes horizontally homogeneous reporting rates, this is particularly problematic in Europe, where occurrence rates are affected by large variations in reporting rates between countries [12]. Several large-scale tornado climatologies exist that are based on tornado reports alone. In addition to those for Europe and the US already discussed, they include Chen et al. [67] for China, Chernokulsky et al. [68] for northern Eurasia and Allen and Allen [69] for Australia, all of which note the potential impact of population bias on their results.

Several studies have examined the impact of population density on tornado reporting. For the US, Jagger et al. [57] and Elsner et al. [58] investigated the influence of orography on tornado occurrence rates in the Great Plains and used population density as an input parameter to locally correct for population bias in a Bayesian framework. For Canada, Cheng et al. (2013) [70] also used a Bayesian approach to create a tornado climatology based on tornado reports, lightning data and population density. They later extended a similar approach to cover the whole of North America [61]. Unlike in the current study, which considers daily meteorological parameters and tornado reports, these studies use seasonal aggregates.

While Cheng et al. (2016) [61] account for population bias on a relatively high spatial resolution of 50 km $\times$ 50 km, we applied a population-bias correction on national level resulting in an estimated tornado occurrence rate of 181 tornadoes per year across the model domain, excluding waterspouts and tornadoes spawned by extratropical cyclones. This is in line with results of an earlier study [62]. Our model domain covers 2.2 million km$^2$ and the average tornado occurrence rate is 82 tornadoes per 100 years and 10,000 km$^2$. This is more than twice as many as 38 tornadoes per 100 years and 10,000 km$^2$ observed in the domain used in Figure 2. Our population-bias correction increased the number of tornadoes per year from 135 to 181, i.e., by 34% indicating that either the tornado density in the non-modelled part of Europe is considerably lower than in the modelled part or reporting biases are higher there. The 1219 tornadoes observed on average per year in the contiguous U.S. represent an average tornado occurrence rate of 160 tornadoes per 100 years and 10,000 km$^2$ which is only twice as high as in our model domain. With large sparsely inhabited areas in North America there might be an even higher population bias than for Europe. Cheng et al. [70] estimated the number of Canadian tornadoes to be more than twice the reported number.

Where reliable observations are not available, information about meteorological conditions from reanalysis data can be used instead. Some convective parameters are specifically designed to forecast the risk of severe weather (such as the Craven parameter) or tornadoes (such as STP). While these parameters are based on studies on U.S. observations, they have been used to create global climatologies [31]. The major advantage of such climatologies is that they are independent of local observations and of reporting biases. The major disadvantage is that using parameters and thresholds created for the U.S. might not produce results that are representative of the local climate elsewhere [71]. Statistical models can be used to combine convection parameters with observations in order to model local tornado occurrence rates. The results, however, depend on the resolution and biases of the model data used to calculate the convection parameters.

Independent of local observation biases and model data, a clear dependence of tornado occurrence rates on orographic features can be seen in the recorded observations as has been shown in Section 2.4. A climatology solely based on orographic features shows several similarities with the results of kernel smoothing and of a logistic regression. Given the strong collinearity between elevation and elevation variability in Europe where no high plains exist, we cannot conclude which orographic feature drives the significant and, in the tail, exponential decrease of tornado occurrence rate with elevation.

RMS has assessed these approaches with the aim of creating a quantitative tornado occurrence climatology to form part of a new Europe Severe Convective Storm (EUSCS) model. This climatology was presented in Section 2.5. It is based on CFSR and ESWD data, careful data cleansing and the combination of a logistic regression with local corrections, both to address limitations of the reanalysis data and to introduce elevation dependence at higher resolution. To our knowledge, no other tornado climatology for Europe exists that provides the same level of detail for such a large region. While Figure 8d is our best view given the existing data and knowledge, it is based on only a short period of observations. With every year of observations added to the ESWD, more can and needs to be learned about tornado occurrence rates to quantify the local tornado risk in Europe.

**Author Contributions:** Conceptualisation, J.G.; methodology, J.G. and P.H.; writing, J.G. and P.H. All authors have read and agreed to the published version of the manuscript.

**Funding:** This research received no external funding.

**Acknowledgments:** The authors thank RMS for providing them with the opportunity to work on this fascinating topic. We thank the three reviewers for their very helpful comments and suggestions.

**Conflicts of Interest:** The authors work for RMS, a company that builds mathematical models of extreme weather events and their impacts. The results of this particular piece of research are used in the RMS EUSCS product.

## Appendix A. Map of the Domain

Figure A1 provides a map of the domain indicating the countries and regions mentioned in the text.

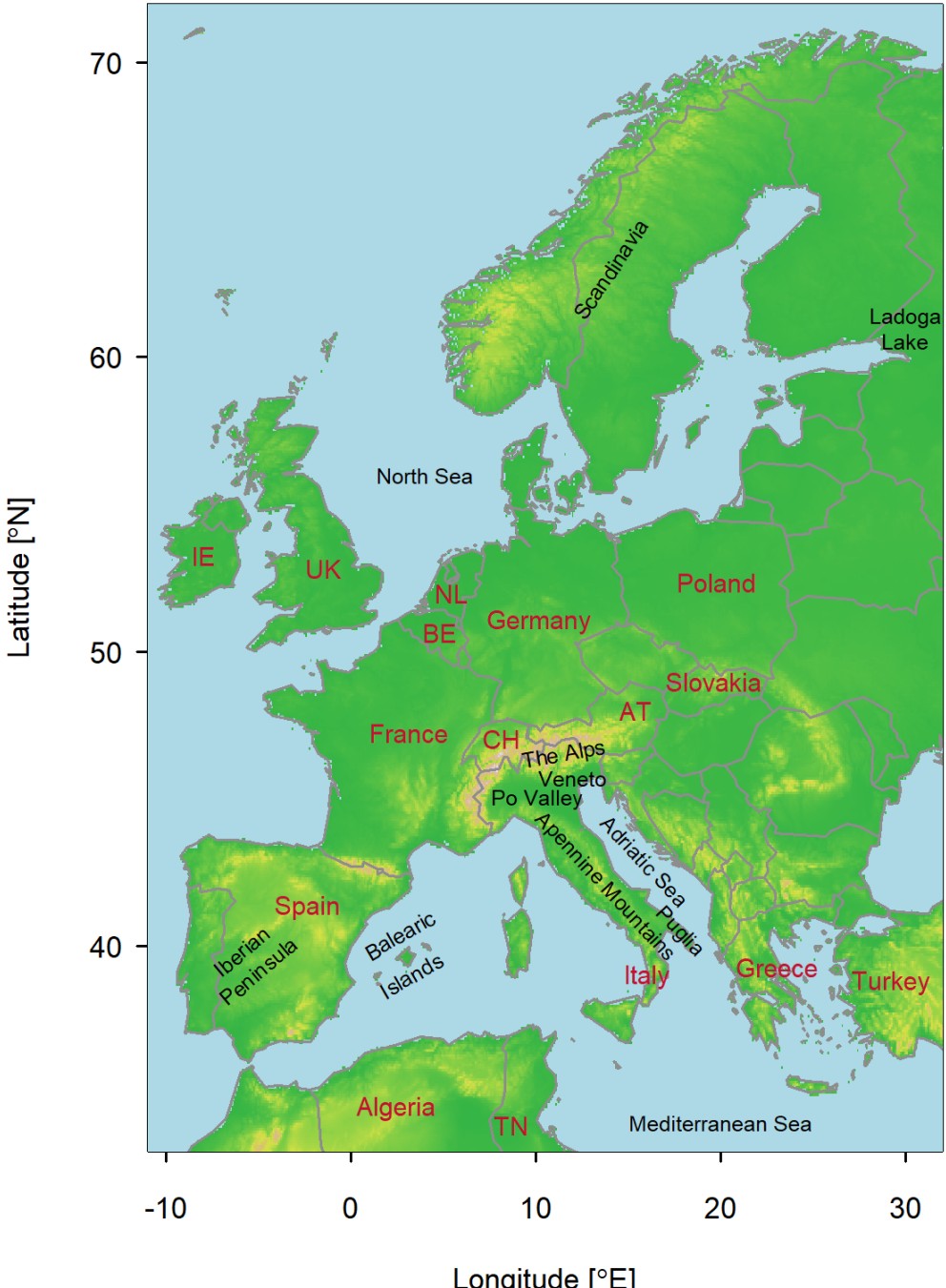

**Figure A1.** Map of the domain. Countries mentioned in the text are indicated either by their name or the two-letter ISO code in red: Austria (AT), Belgium (BE), Switzerland (CH), Ireland (IE), the Netherlands (NL), Tunisia (TN) and United Kingdom (UK). Other regions mentioned in the paper are named as well.

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
