# Peer review of "Tornado Risk Climatology in Europe"

_atmosphere, doi:10.3390/atmos11070768_

Round 1

Reviewer 1 Report

The purpose of the reviewed paper was to construct a quantitative tornado risk map for a significant part of Europe providing number of tornadoes occurring per 100 years and 10,000km2.

Authors presented several different ways to create quantitative maps of tornado occurrence using Kernel smoothing of observations, climatologies of convective parameters from reanalysis, output of a logistic regression model to link convective parameters with observed tornadoes, orography-dependent climatologies and the tornado occurrence rates from the Risk Management Solutions Severe Convective Storms Model. At the end of the paper Authors discussed advantages and disadvantages of each approach and compare the results.

The aim of the research was set correctly. The authors used advanced research methods in its implementation. The obtained research results are interesting and largely systematize knowledge of methods that allow determining the likelihood of tornadoes occurrence in Europe. The work was based on well-selected scientific literature. The paper is suitable for publication after minor comments.

Minor comments:

line 14th of the text – “In the year 1091 a tornado flattened about 600 wooden-frame houses in central London…” The question concerns the year 1091. Is it correct? There is also no source literature provided.

The second part of the description of Figure 1 (Lower panel shows local hit rate per 100,000 years with intensity of at least F3.) is incomprehensible.

Reviewer 2 Report

Tornado Risk Climatology in Europe

The authors present a comprehensive review of quantitative methods used to evaluate tornado risk climatology in Europe. They specifically evaluate kernel smoothing techniques, meteorological condition investigations, statistical (logistic) modeling, and orographic considerations. In total, they find advantages and disadvantages to each approach, which typically are associated with geographic and scale-based decisions.

Overall, I found the article to be well written, informative, and a good review of methods in tornado climatology. I do, however, have some major concerns that I would like to have addressed before I find the manuscript suitable for publication. Many of these concerns are centered around the writing of the manuscript, rather than the methods performed, but I do have some questions pertaining to the standards of tornado occurrence used in this work.

Major Concerns:

(1) I am concerned about the use of registered as a category of tornado throughout the manuscript. As a U.S.-based researcher, we almost unequivocally use the term recorded when discussing tornado that make it into the historical record (Storm Events or the Storm Prediction Center’s record). Is the term registered used in European literature to discuss tornadoes that make the historic record?

(2) Why are tornado occurrence rates broken down per 100 years and per 10,000 square km? The per 10,000 square km definition does appear in the U.S. tornado literature, but none of those works are cited here. The per 100 years definition is not one I am familiar with. Is there a reason this period of time is chosen?

(3) The introductory paragraphs of many of the sections are far too short and distracting to be of any use. Could these individual paragraphs—specifically sections 2.1 and 2.5—be written in a way that draws the reader in more?

Minor Concerns:

I suggest replacing registered with recorded throughout the manuscript.

Line 32: I suggest adding “per” before 10,00 square km

Line 35: I suggest changing the sentence to “… number of tornado occurrences by intensity within a given period.”

Line 53: What does until 2018 mean? Is this inclusive or exclusive of 2018?

Lines 55-56: What is the trend in recorded tornadoes?

Lines 70-72: Recommend adding Elsner, Jagger, and Elsner (2014, PLOS One) to this finding.

Lines 80-81: Can you further explain what you mean by using the results to estimate the local occurrence rate for a hit with given intensity? Is this by F-Scale rating?

Line 82: replace more with greater

Line 82: Why is F3 and greater chosen as the intensity of interest? Add the reasoning here.

Line 89: Replace US$ with USD

Line 91: Replace did with performed

Line 121: What do you mean that no information about quality control of the F-Scale rating for U.S. tornadoes exists? The National Centers of Environmental Information have strict policies related to the classification of tornado magnitude.

Line 128: I suggest “d) only tornadoes recorded since 2005 that touched land and have an intensity rating.”

Line 129: Keep the order of 100 years and 10,000 square km consistent in the text.

Line 135: Are the locations of these tornado occurrences in their proper nouns? For example, should it be written as northern Germany or Northern Germany?

Line 161: Remove “,as in this section”

Line 213: I suggest combing the “Secondly,” and “Lastly,” sentences into one

Line 216: Please provide the logistic regression equation(s). It is hard to follow what the final models included.

Line 216: Add model after logistic regression

Lines 258-260: Is it possible that the nature of reanalysis data (daily maximums) is to blame for the poor performance of CAPE in such models? Please explain this in more depth.

Lines 278-279: Elsner et al. (2016, JAMC) showed a relationship between U.S. tornado occurrence and topography/orography. It might be a good addition to the work cited.

Line 249: Figure 8 could be better described in-text

Reviewer 3 Report

Review - Tornado Risk Climatology in Europe by Grieser and Haines

Recommendation

Major revisions

General Comments

The authors document an interesting study aimed at identifying the tornado risk across most of Europe. The experimental design seems mostly sound and the manuscript is well organized and well written. There are a number of significant problems however that require attention.

The abstract should summarize the results of the study and does not.

Before getting into the various different approaches in Section 2, I'd like to see a discussion of the data i.e. how are events classified as tornadoes (as opposed to downbursts)? What inputs are used? Is the approach similar in each country? Is it similar to the US? Is there a quality control process? Who makes the tornado / no tornado decision? It is essential to know the quality of the data the authors are working with, and each of these questions should be addressed.

There are many locations to which the authors refer in the text, but there is no labeling on a map to know where these locations are. Could use an initial map showing the study area with locations referred to in the text.

There are a number of important assumptions made that are not adequately backed up by the authors, and have a large influence on the results. The authors need to address this, and perhaps rethink parts of the methods used.

There needs to be an additional discussion section that compares the results from the final method with those from other studies in Europe and elsewhere (and even Fig. 1). As it stands, there is little context for the results.

Detailed Comments

Line 1 – How is ‘severe’ defined?

L1 – Should use ‘damage’; ‘damages’ is used specifically for law

L5 – not a sentence

L12 – recommend replacing ‘very’ with ‘potentially’

L12 – omit ‘tornado’ here since it is used a bit later in the sentence

L15 – insert comma after ‘central London’ for clarity

L24 – suggest breaking into two sentences – i.e. ‘…exist. These…’, or change to ‘exist, including…’

L32 – need to say that (or how) this will overcome reporting differences between countries to make this paragraph make sense

L35 – ‘hits’ may be the wrong word here; if used need to explain exactly what that means

L52 – does the SPC collect or just houses the collection? What about Storm Data?

Figure 1 – after studying the two maps carefully, I’ve determined there is no difference between the them (aside from the contours) and there should be. Perhaps the wrong map got used for one of these?

Figure 2 – for these maps and others used in following figures, the land and water need to be different colors in order for the reader to interpret the map correctly

Figure 3 caption – re ‘same as before’, need to be clear what this refers to

L182.5 – The Craven parameter needs a reference

L182.6 – Why make use of SBCAPE rather than MLCAPE for STP? Need to have reason other than SPC using it that way.

L239 – how many is ‘many’? Only two references. Suggest changing that.

L242 – should change ‘to other measures’ to ‘than other measures’

L246 – re ‘further down weights’, need another way to say this that is grammatically correct

L336 – need to know what these ‘local corrections’ are and how they are determined. Objective or subjective process used?

L339 – this assumes that sharp gradients do not actually exist. Is that a safe assumption in Europe with so much terrain and many water bodies? The authors need to explain why this assumption is required.

L342 – I think the reader would benefit from seeing this intermediate step via another map. Maybe a series of small maps to show how the pattern evolves with the application of these steps.

L345 – this is not really a hypothesis but an assumption, and likely not a good one. It is highly unlikely that all tornadoes in Germany area registered. Perhaps all strong tornadoes are registered, but even there it may not be true. The authors need to do a better job of defending this assumption, or use a better estimate of the fraction of tornadoes that are registered in Germany. This affects all of the values shown in Fig. 8 so is important to get right.

L348 – again, would like to see a map showing the results of this intermediate step.

L349 – how does these values compare with other values found in past studies in Europe or elsewhere? What does it tell us about the tornado regime in Europe compared to other locations? This should form a significant part of the discussion section.

L359 – The US as a whole has a varying report rate over its recent history, but the difference in reporting rates are much greater between individual states. Every large country / region is affected by this to some extent. This should be reworded to reflect that.

Round 2

Reviewer 2 Report

The authors have provided a vastly improved manuscript and addressed my major and minor concerns. I believe the work is now suitable for publication with one additional citation.

Line 57: Add Verbout et al. (2006, Wea. Forecasting) to the Doswell and Burgess (1988) citation.

Reviewer 3 Report

Review of Revision 1

The authors have improved the manuscript considerably with this revision. I recommend only minor revisions. Detailed comments are provided below.

With regard to labels on a map, I can’t imagine anyone would be offended by making the manuscript easier to understand, particularly those outside of Europe. I still recommend adding labels on the map for the most relevant locations. At the very least, the names of countries that are mentioned in the text should be labelled.

Comments

L54 - I think there is still some confusion here. My understanding is the NWS forecast offices send local storm reports, and the SPC collects these. The full data set is vetted by NCEI and published in Storm Data. Brooks et al. (2003) states that the data they use are from the Storm Data Archive, and as so are the vetted SPC reports. Please ensure this section is accurate.

Figure 1 - I still find there is a problem with these two identical images (I realize the scales shown are different, but the images are the same). If the top image represents the local occurrence rate of ALL tornadoes, and the bottom image represents the local hit rate for F3+ tornadoes only, there has to be a significant difference in the shaded values, and that is not being shown. In addition, why is the bottom image contoured and the top image is not? The contouring is also not mentioned in the caption.

Figure 3 – Given all four panels in the figure share the same units, the differences between the panels would be illustrated far better by keeping the same color scale for each. For example, the difference between c) and d) is so subtle it appears to be insignificant – but the color scale is considerably different. Please use the same color scale for each.

Line 358 – The statement about population bias needs to be supported by a reference.

Figure 8b - Please circle the two locations on the map.

Line 411 – Please provide the level of detail and region size for the next two tornado climatologies, in order to better show the significance of the work here. For instance, how does the climatology done for Canada compare (see Cheng et al. 2013)?

References

Cheng, V. Y. S., G. B. Arhonditsis, D. M. L. Sills, H. Auld, M. W. Shephard, Wm. A. Gough, and J. Klaassen, 2013: Probability of tornado occurrence across Canada. J. Climate, 26, 9415-9428, DOI 10.1175/JCLI-D-13-00093.1
